# CLUES: Few-Shot Learning Evaluation in Natural Language Understanding

**Subhabrata Mukherjee, Xiaodong Liu, Guoqing Zheng, Saghar Hosseini, Hao Cheng
Greg Yang, Christopher Meek, Ahmed Hassan Awadallah, Jianfeng Gao**
Microsoft Research
{submukhe, xiaodl, zheng, sahoss, chehao}@microsoft.com
{gregyang, meek, hassanam, jfgao}@microsoft.com

## Abstract

Most recent progress in natural language understanding (NLU) has been driven, in part, by benchmarks such as GLUE, SuperGLUE, SQuAD, etc. In fact, many NLU models have now matched or exceeded "human-level" performance on many tasks in these benchmarks. Most of these benchmarks, however, give models access to relatively large amounts of labeled data for training. As such, the models are provided far more data than required by humans to achieve strong performance. That has motivated a line of work that focuses on improving few-shot learning performance of NLU models. However, there is a lack of standardized evaluation benchmarks for few-shot NLU resulting in different experimental settings in different papers. To help accelerate this line of work, we introduce CLUES[1], a benchmark for evaluating the few-shot learning capabilities of NLU models. We demonstrate that while recent models reach human performance when they have access to large amounts of labeled data, there is a huge gap in performance in the few-shot setting for most tasks. We also demonstrate differences between alternative model families and adaptation techniques in the few shot setting. Finally, we discuss several principles and choices in designing the experimental settings for evaluating the *true* few-shot learning performance and suggest a unified standardized approach to few-shot learning evaluation. We aim to encourage research on NLU models that can generalize to new tasks with a small number of examples. Code and data for CLUES are available at https://github.com/microsoft/CLUES.

## 1 Introduction

Benchmarks have provided researchers with well-defined challenges with clear metrics and have driven significant progress on natural language understanding (NLU). In fact, several recent benchmarks such as GLUE [31] and SuperGLUE [30] have made it clear that many current large-scale models can match or exceed "human-level" performance on NLU tasks in these benchmarks, e.g. [6]. Current NLU benchmarks have significant limitations. First, tasks are often limited to those that can be easily represented as classification tasks. Second, and most importantly, there are models that match or exceed "human-level" performance given large amounts of task-specific labeled training data in most of these benchmarks. In contrast, humans can perform complex tasks given only a few demonstrations. These limitations severely undermine claims of achieving broad human-level performance on NLU tasks. In this regard, the CLUES benchmark provides a fair setting to compare machine and human performance given a few training examples across diverse tasks.

We introduce a new few-shot NLU benchmark (CLUES), that aims to address these limitations. Few-shot evaluation of NLU performance has emerged as an important task and is considered to

---

[1]Constrained Language Understanding Evaluation Standard

35th Conference on Neural Information Processing Systems (NeurIPS 2021) Track on Datasets and Benchmarks.

reflect important aspects of human-level language understanding ability. The CLUES benchmark fills the need for a standardized approach to few-shot evaluation and a benchmark to measure progress in *true* few-shot learning [20] while expanding the scope beyond sentence classification tasks.

One of the goals of creating this benchmark is to create a standardized approach to evaluating methods for few-shot learning of NLU tasks. A wide variety of approaches to NLU tasks have emerged; many rely on large pre-trained autoencoding, autoregressive and sequence-to-sequence models. To accommodate different model types and a broader set of tasks beyond sentence classification, we frame all of the tasks in CLUES, including sentence classification tasks, as a 'set of spans' extraction tasks; in which the model outputs a set of text spans[2]. This allows us to provide a novel unified metric across multiple tasks included in the benchmark such as sentence classification, question answering, and named entity recognition.

One of the key criteria for including a task in the CLUES benchmark is that there is a clear gap between human and machine performance. We provide results for both human and machine performance on all tasks. Our human evaluation demonstrates that people are able to perform all tasks at a high level of performance when given only a few labeled examples or even in the zero-shot setting in which they are only given a task description. In order to evaluate machine performance we consider a range of model architectures, a range of model sizes, as well as a set of alternative adaptation techniques. The adaptation techniques include classic full-model fine-tuning, novel task-specific prompt tuning and, in-context learning in the case of GPT-3. While interesting patterns of performance emerged, the key result is that there is a significant gap in performance between current models and human level performance for the tasks in the CLUES benchmark highlighting the need for research to improve few-shot learning for NLU tasks. We hope that our benchmark will encourage NLU research in methods that can learn and generalize to new tasks with a small number of examples.

## 2   Related Work

**Few-shot Learning in NLU** Few-shot learning is the problem of learning a new task with a small number of annotated examples. It has been gaining more traction with advances in large-scale pre-trained language models (e.g.,BERT [3], T5 [23]), which have demonstrated great ability to learn new tasks efficiently. This inspired a line of work on best practices for finetuning pre-trained language models with few labeled samples [41, 26, 37]. GPT models [22, 2] spurred interest in prompt-based or in-context learning, where discrete text prompts are used to condition language models to perform specific tasks. Additional studies explored prompt tuning, where prompts are learned through back propagation using labeled examples [8, 9, 5, 33].

Another line of work explored semi-supervised learning; where unlabeled data, alongside usually small amounts of labeled data, is used for learning [33, 35, 14, 16]. Recent studies have also explored meta-learning in NLU where the models have access to data from many training tasks to learn from, and evaluate the few-shot learning ability on unseen test tasks [4, 1, 18]. In this work, we do not address the meta-learning setting [38]. Rather, our benchmark consists of a carefully chosen set of *fixed* tasks, each with its own (small) training set and test set. The size of the training set is the number of shots, and the model is allowed to adapt to it using various methods, such as classical finetuning, prompt-based finetuning, or GPT-3 style in-context learning.

**NLU Benchmarks** Recent progress in NLU has been driven by the focus on improving performance of benchmark datasets such as MNLI [34] GLUE [31], SuperGLUE [30], SQuAD [25]. For many of these benchmarks, state-of-the-art systems have achieved the best possible performance (often exceeding human-level performance) [6]. However, most these benchmarks assume the model has access to large amounts of manually labeled data. This led to few-shot setting gaining significant interest as an important aspect of measuring NLU performance.

Most work for few-shot learning in NLU uses randomly chosen subsets of existing datasets for evaluation, e.g. [27]. The lack of standard approaches to evaluation and standardized benchmark (with the exception of recently proposed benchmarks for meta-learning evaluation [38]) leads to challenges with estimating the performance of and comparing different few-shot learning approaches [20]. This work aims to bridge this gap.

---

[2]We take inspiration from recent works [23, 15, 7] to unify multiple NLU tasks.

Table 1: Comparing our few-shot NLU benchmark CLUES to contemporary works, where CLUES (1) expands task coverage to include Sequence Tagging in addition to Classification (Class.) and Machine Reading Comprehension (MRC); (2) provides multiple few-shot splits and sizes to measure variance in task performance across splits and shots; (3) provides a unified format for all tasks to eliminate task-specific model customization; (3) provides a single unified metric across all tasks; and (4) provides results for both human and machine performance evaluation (eval.) on all few-shot tasks.

| | CLUES | Few-GLUE | Few-CLUE | Few-NLU |
|---|---|---|---|---|
| Language | English | English | Chinese | English |
| Tasks | Class., MRC Sequence Tagging | Class., MRC | Class., MRC | Class., MRC |
| Few-shot Splits | Multiple | Single | Multiple | Single |
| # Few shots | 10, 20, 30 | 32 | 32 | 64 |
| Unified Format | ✓ | ✗ | ✗ | ✗ |
| Unified Metric | ✓ | ✗ | ✗ | ✗ |
| Few-shot Human Eval. | ✓ | ✗ | ✗ | ✗ |

Table 2: CLUES benchmark design principles.

| Task Selection | Task Formulation | Evaluation |
|---|---|---|
| 1. Significant gap between human and machine performance

2. High coverage and diversity of NLU task types

3. Tasks where context is crucial and factoid knowledge alone is insufficient for answering questions correctly

4. Tasks must be unbiased towards or against any existing class of models | 1. Uniform task format to unify different types of tasks and model families to encourage broad usage and adoption

2. The contexts and questions should be phrased in *unambiguous*, *natural* language

3. Similar to task selection, the questions or prompts should also be model agnostic | 1. Unified metric to compare and aggregate model performance across diverse tasks

2. No separate validation set to mimic a *true* few-shot learning setting

3. Mean and variance across runs on multiple training splits with different random seeds |

We follow recent work that explored unifying the formats of different tasks, in order to facilitate transfer learning especially using large-scale pre-trained language models. For example, DecaNLP [15] processed all tasks into a unified question answering format, UFO-Entail [39] formulated multiple choice QA and co-reference resolution as textual entailment task, and T5 [23] studied unifying all tasks in text-to-text format.

Contemporary to this work, few-shot learning evaluation in NLU has garnered great attention in the research community leading to the development of benchmarks like FewCLUE [36], FewGLUE [27], FewNLU [43] and CrossFit [38] with the following distinctions with our benchmark. CrossFit focuses on multi-task and meta-learning settings where the models have access to data from many training tasks to learn from, in order to evaluate the few-shot learning ability on a new unseen test task. This is different than CLUES which does not address the multi-task setting. Rather, CLUES consists of a carefully chosen set of fixed tasks, each with its own (small) training set and test set. We summarize the difference between CLUES and the other three benchmarks in Table 1.

## 3 CLUES

We seek to provide a standardized evaluation of different few-shot learning approaches and demonstrate a significant gap in the few-shot learning performance between humans and machines for NLU tasks. Our aim is to promote progress in bridging this gap. In particular, our benchmark is intended to evaluate general-purpose models across diverse NLU tasks in few-shot settings. We use the term *general-purpose* to indicate that a single model can be used for all tasks, possibly with task-specific fine-tuning. Note that we do not address the multi-task or cross-task few-shot learning which has been the subject of other studies [38].

**Benchmark Composition** Each task $\mathcal{T} = (td, \mathcal{D}^{Train}, \mathcal{D}^{Test})$ in our collection consists of (a) a natural language task description $td$, (b) training sets $\mathcal{D}^{Train}$ of labeled examples for different

shots, and (c) a test set $\mathcal{D}^{Test}$. Each labeled example consists of a natural language context, a natural language question, and a set of answers (spans) that could also be potentially empty. $\mathcal{D}^{Train}$ for any task contains a total of 30 labeled examples. However, we support benchmarking of 10-shot, 20-shot, and 30-shot performances, for which we organize our training set $\mathcal{D}^{Train}$ into subsets $\mathcal{D}_{10}^{Train} \subseteq \mathcal{D}_{20}^{Train} \subseteq \mathcal{D}_{30}^{Train} = \mathcal{D}^{Train}$, where each $|\mathcal{D}_k^{Train}| = k$. Furthermore, given the variance in few-shot model performance across different seeds and splits of the data, for each $k$-shot setting, we provide 5 training splits (satisfying the subset inclusion criteria above for each split across multiple shots) and a single test set for reporting both the mean and variance in model performance.

## 3.1 Task Selection

We consider the selection of tasks based on the principles outlined in Table 2 with the chosen tasks summarized in Table 3. In what follows we explain our choices and how we applied the principles.

We divide the set of tasks into three distinct categories, namely, classification, sequence labeling and machine reading comprehension to cover a wide spectrum of NLU scenarios. We further unify all of these tasks with a single format by posing them as a 'span extraction' problem (discussed in Section 3.2).

For classification, we focus on both sentence classification and sentence-pair classification. Sentiment Analysis (SA) and Natural Language Inference (NLI) are both popular benchmark tasks. We choose SST-2 [28] for sentiment classification as it poses an interesting challenge given its short context and also as a representative task used in several recent few-shot learning works [5, 16, 13]. For the language inference task, we choose MNLI [34]. Previous work has demonstrated that the performance of different models on the GLUE benchmark [31] tend to correlate with the performance on MNLI, making it a good representative of all tasks in GLUE [21, 11].

Contrary to instance-level classification tasks, sequence labeling is more challenging due to its focus on token-level classification and the dependencies among different tokens. We consider the popular Named Entity Recognition task that aims to identify names of person, organization and location. To this end, we consider both the widely used benchmark task CoNLL03 [29] and the more recently released WikiAnn [19]. These two specific datasets have been extensively studied in prior works in fully supervised settings. We make these tasks more challenging by introducing empty answers (discussed in Section 3.2).

Finally, as the third sub-class of tasks, we consider machine reading comprehension (MRC). MRC tasks require a machine to answer questions based on a given context. This is a challenging task given the requirement of both natural language understanding as well as (commonsense) knowledge reasoning. To this end, we chose one of the most widely used extractive reading comprehension tasks, SQuAD-v2 [24], a standard-bearer reading comprehension dataset created from Wikipedia with manual annotations. The introduction of unanswerable questions makes the task more challenging by preventing simple pattern matching between question and answer sentence. However, it still lacks more sophisticated understanding that re-

Table 3: Task descriptions and statistics.

| Corpus | \|**Train**\| | \|**Test**\| | **Task** | **Domain** |
|---|---|---|---|---|
| Sentence Classification Tasks | | | | |
| SST-2 | 10/20/30 | 210 | SA | reviews |
| MNLI | 10/20/30 | 210 | NLI | misc. |
| Machine Reading Comprehension Tasks | | | | |
| SQuADv2 | 10/20/30 | 200 | QA | Wiki |
| ReCoRD | 10/20/30 | 200 | QA | news |
| Sequence Labeling Tasks | | | | |
| CoNLL03 | 10/20/30 | 600 | NER | news |
| WikiANN | 10/20/30 | 600 | NER | Wiki |

quire reasoning over commonsense knowledge or understanding across multiple sentences in the passage. To further probe a deeper understanding of the machines, we leverage ReCoRD [40] – consisting of curated CNN/DailyMail news articles where queries are filtered out if they are either ambiguous to the human readers or easily solvable by existing MRC systems.

## 3.2 Task Formulation

Following the *Task Formulation* principles in Table 2, we next describe how we sampled and modified examples from available datasets to form our benchmark.

Table 4: Examples of labeled examples in our tasks. We unify all natural language understanding tasks with the format {context, question/prompt, answer} where the answer is given as a *set of spans*. For clarity, we highlight the span(s) in the context and/or question that correspond to each answer.

| Task | Context | Question/Prompt | Answer |
|---|---|---|---|
| SST-2 | The movie was very boring | positive or negative? | {'negative'} |
| MNLI | The Old One always comforted Ca'daan, except today. <SEP> Ca'daan knew the Old One very well. | entail, contradict, or neutral? | {'neutral'} |
| SQuAD | Nikola Tesla (10 July 1856 – 7 January 1943) was a Serbian American inventor | When was Tesla born? | {'10 July 1856'} |
| ReCoRD | The copyright infringement case alleges that the Zeppelin song was taken from the single "Taurus" by the 1960s band | According to claims in the suit, "Parts of 'Stairway to Heaven,' · · · sound almost identical to significant portions of X. What is X? | {'"Taurus"'} |
| CoNLL03 | U.N. official Ekeus heads for Baghdad to meet prime minister Allawi | Set all person names | {'Ekeus', 'Allawi'} |
| WikiANN | He was in private practice in Berks County , Pennsylvania from 1949-1970 . | Set all the locations in the context | {'Berks County' , 'Pennsylvania'} |

**Unifying NLU Tasks with a Single Format**   Pre-trained language models leverage a single base encoder to perform all tasks by adding *task-specific* prediction layers on top of the encoder. This requires different prediction layers for different task formats, for instance, span decoders for question-answering and other MRC tasks, and classification layers for text classification tasks. This further requires different training strategies for different tasks.

In order to address these challenges, we follow and extend recent works [23, 15, 7] to unify all task formats to a *set of spans* extraction task given a question and a context as input, where the set could also be potentially empty. The spans are to be extracted from either the context or the question. While most tasks like MNLI or SQuAD will have unique spans (i.e. set of size 1) as answers, other tasks like CoNLL03 can also have an empty set or a set of more than 1 element as answers. Refer to Table 4 for some illustrative examples.

**Sampling of Training and Test Data**   In this benchmark, we are interested in few-shot learning capabilities and hence we only need enough data to reliably estimate their performance. To this end, we use existing data sets for every task and sample labeled examples to adapt to our setting. In this, we follow similar principles as in [5, 33, 16, 38, 32, 42] to randomly sample labeled examples from the above datasets into $\mathcal{D}^{Train}$ and $\mathcal{D}^{Test}$.

Specifically, for classification tasks, we sample $k \in \{10, 20, 30\}$ labeled examples as few-shot training sets from the available training data for a given task, and $\approx 200$ labeled examples as the held-out evaluation set sampled from the corresponding test data[34]. For NER tasks, we consider a test set of 200 examples for each entity type from {PER, ORG, LOC}. Refer to Table 3 for task statistics. For sequence labeling and machine reading comprehension tasks, we sample $k$ labeled examples for *each question type* corresponding to each entity type for the given task as training examples. For example, the NER task poses three question types of the form *Find the names of all* ENT *in the given context*, where ENT $\in$ {PER, ORG, LOC}. By virtue of such construction, the answer corresponding to some of the entity types for a given context may correspond to empty spans. This makes the task more challenging for models that heavily rely on pattern matching and memorization (e.g., spotting entities encountered during pre-training) and probes the natural language understanding capabilities based on context.

To establish a true few-shot learning setting for this benchmark, **we do not include a separate validation set for any task**. This is to prevent users from using validation sets for training that drastically changes the amount of available supervision and model performance [20] and correspondingly makes comparison of different models difficult. Alternatively, we recommend using a portion of training set

---

[3]MNLI consists of 210 test samples having a balanced distribution over 7 genres with 30 samples each.

[4]One of the objectives of this work is to compare human-machine performance on the same test split for every task. Since it is expensive to obtain multiple human annotations (to measure agreement) on thousands of test examples, we perform sub-sampling to create smaller test sets.

as development set if needed following [20]. Furthermore, to evaluate the effectiveness of additional labeled examples in the few-shot setting, we construct training sets that are subsets of each other.

Given the wide variance in the performance of large pre-trained models in the few-shot setting for different random seeds and training examples [20], we provide *five* different training splits for each shot satisfying the above subset inclusion criteria, such that $\mathcal{D}_{10}^{Train_i} \subset \mathcal{D}_{20}^{Train_i} \subset \mathcal{D}_{30}^{Train_i} : i \in [1, 5]$. This allows us to report both the aggregated model performance and variance across the splits – evaluated on the single test set for each task as provided in this benchmark. The variance can be used as an indicator for model robustness and its stability for few-shot learning.

**Other sampling strategies.** Apart from random sampling, we also considered creating more difficult versions of the tasks via adversarially perturbing context/prompt or by selecting hard questions with respect to a reference model (e.g. BERT or RoBERTa). However, we did not adopt these approaches for the following reasons: (1) We observed that the perturbed examples from such adversarial methods are often unnatural and not readable by humans. (2) Both adversarial perturbation and selection require a reference model, which violates our model-agnostic task formulation principle in Table 2.

### 3.3 Evaluation Metric

We evaluate a model $M$ in the *few-shot* setting with access to the task description along with a few labeled examples $k \in \{10, 20, 30\}$. As we unify all tasks to be span extraction, we devise a unified metric which can be used to evaluate all tasks in our benchmark. Specifically, we devise a metric named *S1*, that computes an instance-based score based on exact string match between elements from the prediction set and the corresponding ground-truth answer set[5] aggregated across all the instances. Formally, given a set of spans for model predictions $\mathbf{p}$, and a set of spans for ground truth answers $\mathbf{a}$ *for one instance*, the per instance *S1* is defined as follows:

$$S1(\mathbf{p}, \mathbf{a}) = \begin{cases} \frac{2}{\frac{1}{p(\mathbf{p},\mathbf{a})} + \frac{1}{r(\mathbf{p},\mathbf{a})}} & \text{if } \mathbf{a} \neq \emptyset, \mathbf{p} \neq \emptyset, p(\mathbf{p},\mathbf{a})r(\mathbf{p},\mathbf{a}) \neq 0 \\ 1 & \text{if } \mathbf{a} = \emptyset, \mathbf{p} = \emptyset \\ 0 & \text{otherwise} \end{cases} \quad (1)$$

where $p(\mathbf{p}, \mathbf{a})$ and $r(\mathbf{p}, \mathbf{a})$ is the precision and recall, respectively defined as $p(\mathbf{p}, \mathbf{a}) = \sum_i 1(\mathbf{p}_i \in \mathbf{a})/|\mathbf{p}|$, $\quad r(\mathbf{p}, \mathbf{a}) = \sum_i 1(\mathbf{p}_i \in \mathbf{a})/|\mathbf{a}|$. For a test set consisting of multiple instances, the overall *S1* score is computed as the average of *S1* scores of all the instances. For classification tasks, the prediction and ground-truth answer sets consist of a single element which makes the *S1* score equivalent to accuracy for such tasks. Throughout this paper we report *S1* scores over all tasks across the benchmark.

## 4 Human Performance

Human performance has been reported on several NLU tasks, however, the annotation methods used to estimate the human performance are not always consistent in how much information about the tasks is provided to the human. Similar to [17], we estimate human performance such that it is consistent across different tasks and is comparable to machine learning models' performance in few-shot settings. We provided non-expert annotators with a few examples and a short task description. In the zero-shot scenario, the annotators didn't receive any examples. We provide the examples of our annotation platform and short task description in Appendix. In the following sections, we explain the data collection and human evaluation processes.

### 4.1 Data Collection Method

We designed an annotation framework on a crowd-sourcing platform to establish human performance on CLUES tasks. For each task, we use 10, 20, and 30 examples from the training set and all of the test set, as used for model training and evaluation. The workers completed a training step (where they were shown the few-shot training examples) and a testing step (where they annotated the test examples) and they were compensated based on an hourly rate ($12/hour). Each example was annotated by three annotators and they were compensated based on the hourly rate to promote fair compensation and high quality annotations.

---

[5]Similar to *F1*, *S1* is derived from precision and recall, but based on sets.

**Training Step** In the training step, for each task we have three scenarios including 10, 20, and 30 examples. Recall that the larger training sets are the super-set of the smaller sets. For each scenario, we recruit three new workers to ensure that the annotators are only exposed to these specific training examples. While annotators are working on the training examples, they receive a short description of the task and after they submit the annotation for each example (from the training set), the correct answer will be revealed to them in a real-time fashion. Our platform does not allow the annotators to change their judgement after seeing the correct answer. Therefore, we can use the training step to filter out annotators whose performance is very low compared to average annotators in the group.

**Annotation Step** In the annotation step, we have four scenarios including the three few-shot scenarios described in training stage and a zero-shot scenario. In the few-shot scenarios, we ask the same group of annotators who worked on the corresponding training examples to work on the test examples. In the zero-shot scenario, we recruit three new judges who have never worked on the task. Note that we collect three annotations from three different workers for each of these four scenarios.

### 4.2 Human Performance Estimates

To calculate human performance, we measure the performance of each annotator and report the mean and standard deviation of three crowd-workers. The human performance on our test set is shown in Table 5. We also present the zero-shot scenario in this table to better understand if human requires training for any of these tasks. SST and ReCoRD tasks demonstrate none or very minimal improvement in few-shot setting compared to zero-shot setting. This implies that human annotators are mostly relying on their own knowledge and the short task description to complete these tasks.

While, on average, human performance tends to improve with more data in the training step for most tasks, we observe that it tends to decline for some tasks when the number of training examples is increased from 20 to 30. This is an interesting and surprising observation and suggests that additional studies are needed to better understand how humans leverage the provided examples and whether there is a point, beyond which, providing more examples could result in no or even negative value. Note that each cell in Table 5 has been annotated by a different set of three annotators and each set of examples used in the training step is a superset of the smaller set (e.g. the 30 shots is a super-set of 20 shots). While this allows us to compare the performance of different annotators in different settings, it does not control for the overall quality of each annotator group, which could be a factor for some of the differences. We provide more analysis of human annotators on the training task in Appendix.

Table 5: Human performance on test set. We report *S1* score and its variance across 3 annotators.

| #Shots | Sentence Classification | | Named Entity Recognition | | Machine Reading Comprehension | |
| --- | --- | --- | --- | --- | --- | --- |
| | SST-2 | MNLI | CoNLL03 | WikiANN | SQuADv2 | ReCoRD |
| 0 | $83.5 \pm 0.6$ | $64.4 \pm 0.6$ | $85.4 \pm 1.8$ | $82.2 \pm 0.4$ | $70.6 \pm 1.0$ | $94.6 \pm 0.5$ |
| 10 | $79.8 \pm 1.2$ | $78.1 \pm 0.2$ | $87.7 \pm 2.0$ | $81.4 \pm 1.1$ | $71.9 \pm 8.0$ | $94.1 \pm 0.5$ |
| 20 | $83.0 \pm 0.5$ | $78.6 \pm 1.7$ | $89.7 \pm 0.4$ | $83.5 \pm 0.1$ | $76.4 \pm 0.5$ | $94.2 \pm 0.8$ |
| 30 | $83.7 \pm 0.6$ | $69.4 \pm 0.8$ | $87.4 \pm 2.1$ | $82.6 \pm 0.4$ | $73.5 \pm 2.0$ | $91.9 \pm 0.2$ |

We also note that our human evaluation results differ from the results in [17] for some of the common tasks. This could be attributed to many reasons including variance in annotator performance or different aggregation settings and metrics. Most notably, in this work, we reported the mean and standard deviation of annotators performance while [17] reported the performance of majority votes. In addition, we are using a different metric ($S_1$ score) as described earlier.

## 5 Results and Discussions

### 5.1 Fine-tuning Strategies

To evaluate the few-shot learning performance, we consider three different representative fine-tuning strategies, recently developed for pre-trained language models (PLMs).

**(a) Classic fine-tuning:** Popularized by [3], classic fine-tuning is a widely-used approach of adapting PLMs for down-stream tasks. It updates both task-specific head and weights from PLMs jointly. Here,

we unify all tasks as **span-extraction** as shown in Table 4. For all considered PLMs, we assume that inputs are prepended with a special token (ST) at the beginning, e.g., ST=[CLS] for BERT. The input text sequence is split by a PLM-specific tokenizer into subword units $w_t, t = 1, \ldots, T$. Then, a PLM takes the sub-word sequence as input to generate the contextualized representations, $\mathbf{h}_1, \ldots, \mathbf{h}_T \in \mathbb{R}^d$, which are the final hidden states from the PLM.

For a span-extraction head, the probability space consists of token positions of target spans. As shown in Table 4, a target span can be found either in the question or in the context. Given a pair of question $q$ and a passage $p$ in the form of "ST [question] [passage]", the PLM produces contextualized embeddings for all input tokens. Specifically, for each token position $t$ in the input, the final hidden vector $\mathbf{h}_t \in \mathbb{R}^d$ is the contextualized token embedding. The span-begin score is computed as $s_b(i) = \mathbf{w}_b^T \mathbf{h}_i$ using a weight vector $\mathbf{w}_b \in \mathbb{R}^d$. The probability for a span start $i$ is $P_b(i) = \frac{\exp(s_b(i))}{Z_b}$, where $Z_b$ is the normalizing factor over all positions. The span-end score $s_e(j)$ and probability $P_e(j)$ are defined similarly. The probability of an answer span $(i, j)$ is $P(i, j) = P_b(i)P_e(j)$. The training is then carried out by maximizing the log-likelihood of the answer span.

**(b) Prompt-based fine-tuning:** Due to the gap between pre-training and task objectives, the few-shot setting is particularly challenging for classic fine-tuning, where the limited labeled data is inadequate for adapting the task-specific head and PLM weights effectively. Prompt-based fine-tuning addresses this gap, by formulating the task objective in a format as close to the pretraining objective as possible. It directly leverages the pre-trained (masked) language model as the task head, without introducing additional parameters, and has been shown to outperform classic fine-tuning on several few-shot natural language understanding and generation tasks [5, 9, 8, 10]. Here, we adopt the same set of pattern templates and verbalizers as in [5] for SST-2 and MNLI with different PLMs. We refer interested readers to the above work for details. For NER and MRC with diverse output space, it is quite complicated to adapt prompt-based fine-tuning, and we thus defer that to future work.

**(c) GPT-3 in-context learning:** In addition, we conduct evaluations of in-context learning by directly querying GPT-3 without any parameter update. Prediction results are obtained via the GPT-3 API with $k$ labeled examples as demonstrations for each example in the test set. We construct the input context by using the labeled data as examples and feeding them to the API for prediction.

## 5.2 Analysis of Results

In the following, we evaluate the performance of representative state-of-the-art PLMs with different adaptation strategies as discussed above. First, we compare the performance between few-shot and fully supervised settings in our benchmark for different PLMs with varying sizes. Here, we include 5 PLMs from different model families, i.e., auto-encoding masked LM (BERT [3], RoBERTa [12], DeBERTa [6]), auto-regressive LM (GPT-3 [2]) and sequence-to-sequence (T5 [23]). Since the task-specific layers of the models add few parameters as compared to the PLM encoder, we only report the original model capacity in this paper.

For each task, we report macro-averaged results for each model trained on five different splits and evaluated on the corresponding test split along with the standard deviation. The results are summarized in Table 6 for classification tasks, and Table 7 for NER and MRC, respectively.

**Fine-tuning strategies:** For classification tasks (SST-2 and MNLI), we find that prompt-based fine-tuning significantly outperforms its classic fine-tuning counterpart across the board. However, this advantage disappears in the fully supervised setting where both strategies perform similarly. In addition, GPT-3 in-context learning is very effective for SST-2, surpassing all few-shot training baselines (both classic and prompt-based strategies) and almost matching human performance. In contrast, GPT-3 in-context learning produces random guesses for MNLI, indicating the impact of task difficulty on few-shot learning. For both NER and MRC tasks, it is complicated to adapt the current prompt-based approaches. Vanilla adaptation of prompt-tuning for token-level classification with a non-generative model by predicting the label of each token in a sentence, one at a time, yielded close to random performance. However, given its promising results in classification, it is an interesting future direction for designing new prompting mechanisms for such tasks. Additionally, the lengthy input prohibits the adoption of in-context learning with GPT-3 for these task types as well.

**Model capacity:** In the fully supervised setting with adequate training data, the performance of different models generally increase with increasing model size. However, for the few-shot setting, we do not observe any consistent trend or impact of the model size on the performance with classic

Table 6: Performance comparison of humans vs. PLMs on few-shot text classification. `FT`, `PT` and `ICL` stand for classic fine-tuning, prompt-based fine-tuning and in-context learning, respectively. Model variance is reported across five splits for each setting.

| | | SST-2 | | | | MNLI | | | |
|---|---|---|---|---|---|---|---|---|---|
| Shots (K) | | 10 | 20 | 30 | All | 10 | 20 | 30 | All |
| Human | | 79.8 | 83.0 | 83.7 | - | 78.1 | 78.57 | 69.4 | |
| BERT$_{Base}$ | FT | 46.2 (5.6) | 54.0 (2.8) | 53.6 (5.5) | 98.1 | 37.0 (5.2) | 35.2 (2.7) | 35.4 (3.2) | 81.6 |
| (110M) | PT | 63.9 (10.0) | 76.7 (6.6) | 79.4 (5.6) | 91.9 | 40.4 (1.8) | 42.1 (4.4) | 42.5 (3.2) | 81.0 |
| BERT$_{Large}$ | FT | 46.3 (5.5) | 55.5 (3.4) | 55.4 (2.5) | 99.1 | 33.7 (0.4) | 28.2 (14.8) | 33.3 (1.4) | 80.9 |
| (336M) | PT | 63.2 (11.3) | 78.2 (9.9) | 82.7 (4.1) | 91.0 | 41.7 (1.0) | 43.7 (2.1) | 45.3 (2.0) | 81.9 |
| RoBERTa$_{Large}$ | FT | 38.4 (21.7) | 52.3 (5.6) | 53.2 (5.6) | 98.6 | 34.3 (2.8) | 33.4 (0.9) | 34.0 (1.1) | 85.5 |
| (355M) | PT | 88.8 (3.9) | 89.0 (1.1) | 90.2 (1.8) | 93.8 | 57.7 (3.6) | 58.6 (2.9) | 61.6 (3.5) | 87.1 |
| DeBERTa$_{Large}$ | FT | 43.0 (11.9) | 40.8 (22.6) | 47.7 (9.0) | 100.0 | 27.4 (14.1) | 33.6 (2.5) | 26.7 (11.0) | 87.6 |
| (400M) | PT | 83.4 (5.3) | 87.8 (3.5) | 88.4 (3.3) | 91.9 | 44.5 (8.2) | 60.7 (5.3) | 62.9 (3.1) | 88.1 |
| T5$_{Large}$ (770M) | FT | 51.2 (1.8) | 53.4 (3.2) | 52.3 (2.9) | 97.6 | 39.8 (3.3) | 37.9 (4.3) | 36.8 (3.8) | 85.9 |
| GPT-3 (175B) | ICL | 85.9 (3.7) | 92.0 (0.7) | 91.0 (1.6) | - | 33.5 (0.7) | 33.1 (0.3) | 33.2 (0.2) | - |

fine-tuning for most tasks. However, for the two tasks that prompt tuning is used for (SST-2 and MNLI), bigger models tend to perform better.

**Training labels:** There is a significant performance gap between few-shot and fully supervised settings. For classic fine-tuning, there is no consistent trend of performance improvement with a few added training examples; whereas a limited additional number of labeled examples can improve the model performance with prompt-based fine-tuning – suggesting that the latter method is more effective in leveraging additional labeled examples for the few-shot setting.

**Model variance:** For classic fine-tuning, bigger models are observed to have significantly higher performance variance over different training splits, with BERT$_{Base}$ (the smallest model considered) exhibiting the least variance across all tasks[6]. Interestingly, for prompt-based fine-tuning, larger models have less variance as they are likely to learn more effectively with pre-trained language modeling head. However, DeBERTa and T5 are exceptions which can be partially attributed to the difference in the pre-training strategy and the corpus.

**Task difficulty:** For a simple task like SST-2, few-shot performances with prompt-based tuning and in-context learning with GPT-3 are very competitive, and close to (or even better than) human performance. In contrast, for more complex tasks like NER and MRC, most of the pre-trained models with varying sizes obtain close to random performance. Therefore, it is very important to develop more effective few-shot learning methods for such tasks.

**Model vs. human performance:** In the fully supervised setting, all the models exceed human performance substantially for all considered tasks. However, in the few-shot setting, there is a huge gap between the model performance and that of the humans. The only exception is SST-2 where few-shot GPT-3 outperforms humans. We still retain this task as we observe significant few-shot performance gap between humans and all other models. Furthermore, this gap is more pronounced for more complex tasks like NER and MRC where humans perform very well with only a few demonstrative examples whereas all the PLMs perform close to random.

## 6 Conclusion and Future Work

This work has been motivated by the lack of standardized benchmarks and principles to evaluate few-shot NLU models. More importantly, this benchmark has been designed for a fair comparison between human and machine performance on diverse NLU tasks given a few demonstrative examples.

---

[6]For complex tasks like MRC and sequence tagging, BERT-Base often generates empty lists as answers when trained with only a few examples. The default accuracy indicates the proportion of empty lists (questions that do not have answers) in our test sets, resulting in zero variance in some cases.

Table 7: Performance comparison of humans vs. PLMs on few-shot benchmark for NER (CoNLL03 and WikiAnn) and MRC (SQuAD and ReCoRD). Only standard fine-tuning performance is reported along with model variance across five splits for each setting (GPT-3 results discussed in Section 5.2).

| | CoNLL03 | | | | WikiANN | | | |
|---|---|---|---|---|---|---|---|---|
| Shots (K) | 10 | 20 | 30 | All | 10 | 20 | 30 | All |
| Human | 87.7 | 89.7 | 87.4 | - | 81.4 | 83.5 | 82.6 | - |
| $BERT_{Base}$ | 51.3 (0) | 51.3 (0) | 51.3 (0) | 89.3 | 62.8 (0) | 62.8 (0) | 62.8 (0) | 88.8 |
| $BERT_{Large}$ | 51.3 (0) | 51.3 (0) | 51.3 (0) | 89.8 | 62.8 (0) | 62.6 (0.4) | 62.5 (0.6) | 91.0 |
| $RoBERTa_{Large}$ | 50.8 (0.5) | 44.6 (5.1) | 44.7 (2.6) | 93.2 | 58.5 (8.8) | 56.9 (3.4) | 48.4 (6.7) | 91.2 |
| $DeBERTa_{Large}$ | 50.1 (1.2) | 47.8 (2.5) | 48.2 (2.9) | 93.6 | 58.5 (3.3) | 57.9 (5.8) | 58.3 (6.2) | 91.1 |
| $T5_{Large}$ | 46.3 (6.9) | 50.0 (0.7) | 51.2 (0.1) | 92.2 | 61.7 (0.7) | 62.1 (0.2) | 62.4 (0.6) | 87.4 |
| | SQuAD v2 | | | | ReCoRD | | | |
| Shots (K) | 10 | 20 | 30 | All | 10 | 20 | 30 | All |
| Human | 71.9 | 76.4 | 73.5 | - | 94.1 | 94.2 | 91.9 | - |
| $BERT_{Base}$ | 46.0 (2.4) | 34.9 (9.0) | 32.6 (5.8) | 76.3 | 10.3 (1.8) | 11.7 (2.4) | 13.1 (3.3) | 54.4 |
| $BERT_{Large}$ | 42.3 (5.6) | 35.8 (9.7) | 35.3 (6.4) | 81.8 | 9.9 (5.2) | 11.8 (4.9) | 14.9 (3.4) | 66.0 |
| $RoBERTa_{Large}$ | 38.1 (7.2) | 40.1 (6.4) | 43.5 (4.4) | 89.4 | 12.0 (1.9) | 9.9 (6.2) | 16.0 (2.8) | 80.3 |
| $DeBERTa_{Large}$ | 41.4 (7.3) | 44.4 (4.5) | 38.7 (7.4) | 90.0 | 15.7 (5.0) | 16.8 (5.7) | 21.1 (3.6) | 80.7 |
| $T5_{Large}$ | 43.6 (3.5) | 28.7 (13.0) | 43.7 (2.7) | 87.2 | 11.9 (2.7) | 11.7 (1.5) | 12.0 (3.8) | 77.3 |

Recent studies demonstrate several issues in evaluating *true* few-shot learning including the usage of additional held-out examples for tuning hyper-parameters, prompts and templates, and the high variance in the model performance given the choice of seeds and few-shot training examples. To mitigate these issues for training and evaluating few-shot models, the CLUES benchmark adopts and demonstrates the impact of the following design principles.

**Variance matters.** We provide five different splits with different seeds for $k \in \{10, 20, 30\}$ training examples and a single test set to measure the robustness and generalizability of large language models. We observe a wide variance in the few-shot performance with classic fine-tuning that is exacerbated by the model size (refer to Appendix), although the impact is less on prompt-based fine-tuning.

**Validation matters.** We do *not* provide additional validation examples to preserve the *true* few-shot nature of the tasks following [20]. As an artefact of this choice, we train every model for a fixed number of epochs and learning rate. In order to demonstrate the impact of validation set on few-shot performance, we perform a simple experiment. We fix the number of shots as $K = 10$ and the base encoder as BERT-base. We use one of the five training splits as held-out validation set. We train the model on each of the four remaining splits while selecting the best model for each split based on validation loss. We observe the average performance of these models on our test set for SST-2 to be 7% higher than that reported in Table 6 for classic fine-tuning without using any validation set.

**Task difficulty matters.** While prior few-shot learning works primarily explore instance classification tasks to demonstrate few-shot learning capabilities of large language models, the CLUES benchmark incorporates diverse structured classification and reading comprehension tasks. As the complexity of the tasks increase, we observe significantly larger gaps in the few-shot model performance compared to both the fully supervised and human performance.

In this work, our focus has been limited to natural language understanding, where we provide humans and machines with only text information for performance comparison. While humans acquire knowledge from diverse modalities including visual cues and natural language, the pre-trained language models only have access to text information. Therefore, a natural extension of this work is to benchmark few-shot learning capabilities of models and machines in multi-modal settings.

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
