# OpenReview forum: "Few-Shot Learning Evaluation in Natural Language Understanding"
_NeurIPS.cc/2021/Track/Datasets_and_Benchmarks/Round2 — NeurIPS 2021 Datasets and Benchmarks Track (Round 2)_

### Official Review · Reviewer_UQnn · 2021-09-07
**A Few-Shot Learning Benchmark for Natural Language Understanding.**

**Rating:** 6
**Confidence:** 4
**Correctness:** The data collection and evaluation me…
**Clarity:** This paper is well written and easy t…

**Strengths:**

- The authors use a unified task formulation and propose an evaluation metric (S1) for evaluating all the tasks in CLUES.
- The authors have done a good job implementing multiple baselines with different language models and few-shot strategies.


**Weaknesses:**

- The authors should have made some direct comparisons with the existing few shot NLU benchmarks (FewGLUE [1], CROSSFIT [2], and FewCLUE [3]) to highlight the advantages of CLUES.
- There could be an error analysis to help readers understand the limitations of current SOTA few-shot models in the CLUES benchmark.
- The implementation of baselines is not publicly available.


[1] It’s not just size that matters: Small language models are also few-shot learner. NAACL 2021
[2] CROSSFIT : A Few-shot Learning Challenge for Cross-task Generalization in NLP. 2021
[3] FewCLUE: A Chinese Few-shot Learning Evaluation Benchmark. 2021

**Additional Feedback:**

None

**Documentation:**

The documentation of this benchmark is not well done. The maintenance plan, baseline implementation, and evaluation scripts are missing.

**Ethics:**

The authors have discussed the broader impact of this work. I do not have additional concerns.

**Relation To Prior Work:**

As mentioned in weakness, authors could make a table and list all the few-shot NLU benchmarks and highlight the advantages of CLUES.

**Summary And Contributions:**

This paper proposes CLUES, a benchmark for evaluating the few-shot learning capabilities of NLU models. CLUES includes two sentence classification tasks (SST-2 and MNLI), two machine reading comprehension tasks (SQuADv2 and ReCoRD), and two sequence labeling tasks (CoNLL03 and WikiANN), where each task contains 10 to 30 samples for training. All the tasks are formulated as extractive QA problems and evaluated under a true few-shot setting (no extra validation set available). The authors applied multiple pre-trained language models, such as BERT, RoBERTa, T5, GPT3, with 3 different few-shot strategies (fine-tuning, prompt-based fine-tuning, and in-context learning). Experimental results show substantial gaps between the best model and human baseline in most of the tasks.

---

> ### Author Response · Authors · 2021-09-28
> **Author Response**
>
> We thank the reviewer for the helpful feedback. Please find our response below.
>
>
> ### 1. Related Work on Few-shot Learning in NLU
>
> CROSSFIT focuses on multi-task and meta-learning settings where the models have access to data from many training tasks to learn from, in order to evaluate the few-shot learning ability on new unseen test task. This is different than CLUES which does not address the multi-task setting. Rather, CLUES consists of a carefully chosen set of fixed tasks, each with its own (small) training set and test set.
>
> Few-GLUE and Few-CLUE were both created by sampling 32 instances from each task in GLUE and CLUE respectively. We summarize the difference between CLUES and the other two datasets below:
>
> |                                        | CLUES                                     | Few-Glue        | Few-Clue             |
> |----------------------------------------|-------------------------------------------|-----------------|----------------------|
> | Language                               | English                                   | English         | Chinese              |
> | Tasks                                  | Classification, MRC and Sequence tagging  | Classification  | Classification, MRC  |
> | Few-shot Splits                        | Multiple                                  | Single          | Multiple             |
> | # few shots                            | 10, 20, 30                                | 32              | 32                   |
> | Unified Format                         | Yes                                       | No              | No                   |
> | Unified Metric                         | Yes                                       | No              | No                   |
> | Few-shot Human Performance Evaluation  | Yes                                       | No              | No                 |
>
> In summary, CLUES: (1) expands task coverage to include MRC and sequence tagging in addition to classification, (2) provides multiple few-shot splits and sizes, (3) provides a unified format for all tasks to eliminate task-specific model customization, (3) provides a single unified metric across all tasks and (4) provides results for both human and machine performance on all few-shot tasks
>
> ### 2. Error Analysis
>
> Thanks for the suggestion. We will consider this for future work.
>
> ### 3. Implementation of Baselines
>
> The implementation of all baselines, evaluation scripts, sampling and data processing scripts etc. will be made publicly available on Github. The code and data are available for review in the following link: https://aka.ms/CLUES-Code
>
> The data has also been made available during the review process in the submitted URL.
>
>
> ### 4. Maintenance Plan
>
> We will maintain a leaderboard in our Github page, allowing researchers to submit their results as entries.
>
> #### Submission Requirements:
>
> - Each submission must be submitted as a pull request modifying the markdown file underlying the leaderboard.
>
> - The submission must attach an accompanying public paper and public source code for reproducing their results on our dataset.
>
> - A submission can be towards any subset of tasks in our benchmark, or towards the aggregate leaderboard.
>
> - For any task targeted by the submission, we require evaluation on all 5 splits of the corresponding dataset and a report of their mean and standard deviation.
>
> #### Maintaining and Evaluating Submissions: We will check each submission, to the best of our ability, that:
>
> - The model did not pre-train directly on our dataset.
>
> - The submission does not use external data or data from other splits during few-shot fine-tuning, either as extra training set or as validation set for hyperparameter tuning.
>
> - There are no other obvious errors. However, we will not evaluate the merit of the submission’s technical approach.
>
> - We will attempt to reproduce a submission’s results only if we believe there is an error or the submission is made in bad faith.

---

> > ### Comment · Reviewer_UQnn · 2021-09-29
> > **Review Update**
> >
> > Thank you for your response. I have updated my rating accordingly.

---

### Official Review · Reviewer_RWYe · 2021-09-19
**A promising benchmark for few-shot NLU**

**Rating:** 7
**Confidence:** 3
**Clarity:** The paper is easy to read and follow.

**Strengths:**

1. This paper provides a standardized benchmark for evaluating different few-shot learning approaches.
2. Results for both humans and models with different adaption techniques are further provided. The proposed benchmark may facilitate progress in bridging the relatively large performance gap between humans and machines.
3. The design principles of the benchmark including validation, variance and task difficulty are relatively sound.


**Weaknesses:**

1. For subsection 3.1 Task selection, I do not see the explicit connections between the design principles (Table 1) and the chosen tasks.  For example, which principle is considered when choosing CoNLL03 and WikiANN as the tasks. Why not other sequence labelling tasks?
2. For the related work section, the paragraph on few-shot learning in NLU can be extended a bit to give a more comprehensive review of recent efforts on few-shot learning and why such a standardized benchmark is needed. As mentioned by the other reviewer, I will also suggest the author provide more detailed comparisons with the existing few shot NLU benchmarks (FewGLUE [1], and FewCLUE [2]). Based on the existing submission, I can see the differences between CLUES and CROSSFIT.
3. The implementation of baselines is not publicly available.

[1] It’s not just size that matters: Small language models are also few-shot learner. NAACL 2021
[2] FewCLUE: A Chinese Few-shot Learning Evaluation Benchmark. 2021


**Additional Feedback:**

No

**Correctness:**

The construction of the benchmark is relatively sound given their goals. The experimental settings and baselines they established for their benchmark seem sound.

**Documentation:**

The construction of the benchmark is quite well documented. The URL to the benchmark is provided.

**Ethics:**

No ethical concerns.

**Relation To Prior Work:**

The related work section gives a brief introduction of few-shot learning in NLU, and an overview of existing NLU benchmarks.

**Summary And Contributions:**

Most of the existing benchmarks for NLU give models access to relatively large amounts of labelled data for training. This paper introduces a
standardized benchmark and principles to evaluate few-shot NLU models. This benchmark has been designed for a relatively fair comparison between human and machine performance on three types of NLU tasks including sentence classification, sequence labelling and machine reading comprehension. Different pre-trained language models and adaptation methods have been evaluated on the proposed benchmark.

---

> ### Author Response · Authors · 2021-09-28
> **Author Response**
>
> We thank the reviewer for the helpful feedback. Please find our response below.
>
> ### 1. Principles behind choosing CoNLL03 and WikiANN
>
> All principles in the task selection column apply as follows: (1) these tasks exhibit significant performance gap between SOTA models and humans in few-shot settings; (2) sequence labelling tasks have not been included in previous NLU benchmarks; (3) context is crucial to answer, especially with the introduction of empty answers; (4) they have no bias towards any family of models. These two specific datasets have been chosen because they are representative sequence labeling tasks that have been extensively studied in prior works in fully supervised settings. These principles were also used to exclude some tasks (refer to Appendix A Lines 3-10).
>
> ### 2. Related Work on Few-shot Learning in NLU
>
> CROSSFIT focuses on multi-task and meta-learning settings where the models have access to data from many training tasks to learn from, in order to evaluate the few-shot learning ability on new unseen test task. This is different than CLUES which does not address the multi-task setting. Rather, CLUES consists of a carefully chosen set of fixed tasks, each with its own (small) training set and test set.
>
> Few-GLUE and Few-CLUE were both created by sampling 32 instances from each task in GLUE and CLUE respectively. We summarize the difference between CLUES and the other two datasets below:
>
> |                                        | CLUES                                     | Few-Glue        | Few-Clue             |
> |----------------------------------------|-------------------------------------------|-----------------|----------------------|
> | Language                               | English                                   | English         | Chinese              |
> | Tasks                                  | Classification, MRC and Sequence tagging  | Classification  | Classification, MRC  |
> | Few-shot Splits                        | Multiple                                  | Single          | Multiple             |
> | # few shots                            | 10, 20, 30                                | 32              | 32                   |
> | Unified Format                         | Yes                                       | No              | No                   |
> | Unified Metric                         | Yes                                       | No              | No                   |
> | Few-shot Human Performance Evaluation  | Yes                                       | No              | No                 |
>
> In summary, CLUES: (1) expands task coverage to include MRC and sequence tagging in addition to classification, (2) provides multiple few-shot splits and sizes, (3) provides a unified format for all tasks to eliminate task-specific model customization, (3) provides a single unified metric across all tasks and (4) provides results for both human and machine performance on all few-shot tasks
>
> ### 3. Implementation of Baselines
>
> The implementation of all baselines, evaluation scripts, sampling and data processing scripts etc. will be made publicly available on Github. The code and data are available in the following link: https://aka.ms/CLUES-Code
>
> The data has also been made available during the review process in the submitted URL.

---

> > ### Comment · Reviewer_RWYe · 2021-10-03
> > **Review Update**
> >
> > Thanks for your detailed reply, which addressed my concerns. I would like to keep the rating as a good paper, accept.

---

### Official Review · Reviewer_paVL · 2021-09-20
**A benchmark for evaluating the few-shot learning capabilities of NLU models**

**Rating:** 7
**Confidence:** 3

**Strengths:**

1. The motivation of the work is valid. Since the human and pre-trained language model both have general-domain "knowledge", it is unknown how they perform differently in the few-shot learning NLU tasks.


2. The provided benchmark (CLUES) provides a fair setting to compare machine and human performance given a few training examples across diverse tasks. The authors also designed an annotation framework to measure the human performance on CLUES tasks and make the experiment reliable.


3. They conducted a large number of experiments considering the model families and fine-tuning strategies, and also analyzed the influence of different strategies on the few-shot learning ability of pre-train language models. It may enlighten future research.


**Weaknesses:**

1. The comparison experiment is not complete. As mentioned in Section.5.1.(b), the authors did not design the prompt-based approaches for NER and MRC tasks. Though the performance gap is large between human and machine in these tasks, the experiments must be consistent enough.


2. One minor weakness of the motivation: Humans learned the knowledge in a multimodal way including visual clues and text information, but the pre-trained language only pre-trained on the text information. If we compare the few-shot learning ability, it may not
be fair from this perspective.


3. The creation details of the datasets are not clear. See "Additional Feedback" parts.


**Additional Feedback:**

I have some questions for the authors:

1. What is the sampling method of training and test data? Did you claim the seed of the random sampling method? Do you have a better way of splitting the data if we want to make the sampled data more challenging/discriminative.


2. Are the educational background and the first language of annotators the same?


3. For tuning the model in a task-specific way, some models need to add extra layers. It will increase the size of model parameters. Which type of model capacity did you report in Table 5? Original or Modified models?

There are some typos in the paper:
- Line 42: spans.^2. -> spans^2.
- Line 81: exiting -> existing



**Clarity:**

The paper is well written and organized except for:
1. Section.5, the best results in the tables should use **bold** font.
2. The symbols of S1 metric are not consistent. Should we use *S1* or $S1$ ?


**Correctness:**

The benchmark construction and the evaluation method look sound. But in line 202, the authors mentioned that:
> … the overall *S1* is computed as the average of *S1* of all the instances ...

*S1* is a metric, not a numerical value. I suggest that using the following expression:

> … the overall *S1* score is computed as the average of *S1* scores of all the instances ...


**Documentation:**

The datasets are available with clear documentation and license. The authors provided a private link hosted by Onedrive platform.


**Ethics:**

I did not see any ethical problems. The data was annotated by the crowd-sourced annotators. The annotators were well paid with a clear price.


**Relation To Prior Work:**

The discussion of related work is detailed with appropriate references of previous work.

**Summary And Contributions:**

This paper proposes a benchmark for few-shot learning in natural language understanding (NLU) called CLUES, taking the gap between human and machine learning into consideration. The motivation behind creating this benchmark is that the NLU models generally are trained/fine-tuned with large-scale task-specific data whereas the human can generalize the learned knowledge to these tasks with few examples. The main idea is also straightforward: sample a few-shot training set then ask human annotators to simulate the few-shot learning process using this dataset. Finally, the authors compare the machine and the human performance, and the results reveal that there exist large performance gaps between human and machine in the few-shot learning tasks regardless of fine-tuning strategies, model families, etc.

The authors not only contribute a new few-shot learning benchmark for NLU involving human performance, but also provide the detailed comparison and preliminary analysis in terms of the performance gap.

---

> ### Author Response · Authors · 2021-09-28
> **Author Response**
>
> We thank the reviewer for the helpful feedback. Please find our response below.
>
> ### 1. Prompt-based Approaches for NER and MRC Tasks
>
> To the best of our knowledge, prompt-based fine-tuning has not shown similar success with sequence labeling tasks as with instance-level classification or generation tasks. To further study this, we tested prompt-based fine-tuning on both NER and MRC tasks. For instance, NER tasks require a label for each token. To accommodate token-level classification with prompt-based fine-tuning and a non-generative model, we tried to predict the label of each token in a sentence one at a time. This not only resulted in a significantly higher cost for training/inference, but also much longer sequence lengths for demonstrations (exceeding 512 sequence length in many cases). Even with trimming the input sequence length, prompt-based fine-tuning yielded close to random performance. For instance, in our experiments, prompt-tuning achieved an S1 of 51.1-51.4 on CoNLL2003 and 62.8 on WikiANN (a naïve prediction of all empty lists yields an S1 of 51.3 and 62.8 respectively for CoNLL and WikiANN), an S1 of almost 0 for both SQuAD 2.0 and ReCoRD. This demonstrates the need for additional research and methodological innovation to support prompt-tuning for sequence labeling tasks which is beyond the scope of this paper.
>
>
> ### 2. Multimodal Learning
>
> While we agree that multimodal understanding is an interesting direction, our focus in this work was limited to natural language understanding rather than multimodal understanding. As such, in our human evaluation, we consider a closed-world assumption and provide the individuals with only the textual information for every task.
>
> ### 3. Dataset Creation Details
>
> We use a different random seed to sample different splits of our training data (there are 5 splits for every task) from the original training data of the task. Within the same split, we further sample different shots so as to ensure $D_{10}^{Train_i} \subset D_{20}^{Train_i} \subset D_{30}^{Train_i}:\ i \in [1, 5]$. The test data (single split) is also randomly obtained from the original test set by fixing the seed. All seeds are provided with the code.
>
> We also considered creating more difficult versions of the tasks via adversarially perturbing context/prompt or by selecting hard questions w.r.t. a reference model (e.g. BERT or RoBERTa). However, we ultimately did not adopt these approaches for the following reasons (also mentioned in Appendix A, Lines 11-16): (1) We observed that the perturbed examples from such adversarial methods are unnatural and typically not readable by humans. (2) Both adversarial perturbation and selection require a reference model, which violates our model-agnostic task formulation principle in Table 1, and (3) there is a significant gap between human and machine performance for the few-shot setting even with the simpler random sampling strategy.
>
> ### 4. Human Annotators Background
>
> We did not collect any personal information from the annotators, but all the annotators were required to speak English fluently.
>
> Additionally, we have a training step for the annotators (Section 4.1 lines 222-229) where we filter out annotators whose performance is very low compared to average annotators in the group.
>
> ### 5. Task-specific Layer Size
>
> Task-specific layer only contains around 2K trainable parameters which are very few compared to the pre-trained model parameters. Thus, we reported the original model capacity in paper.

---

> > ### Comment · Reviewer_paVL · 2021-09-29
> > **Reply to the authors**
> >
> > The authors address my concerns regarding the prompt-based fine-tuning, dataset/model details and human annotation. I also would encourage the authors to fix them in the camera-ready version.
> >
> > Thanks for the reply. I would like to raise my rating of this paper.

---

### Official Review · Reviewer_LHkf · 2021-09-20
**Very well researched, structured and presented study with strong impact and application**

**Rating:** 8
**Confidence:** 4
**Clarity:** OK

**Strengths:**

- The subject is very well explained and important for the field.
- The paper is well-written.
- Six tasks are chosen from different aspects.

**Weaknesses:**

- In many datasets, there exist a separate test set. Are the results reported based on evaluation on those test sets or an average on randomly split sets? If the latter, please explain why not to use the already existing standard test sets.

- In Table 6, the validation report for the BERT model in CoNLL (and WikiANN) is the same for all values of K. In addition, the variance is 0 for 5 validation splits. Is there a specific reason for that, considering that other BERT-based approaches differ in values.

- In the first task, values less than 50 are reported for DeBERTa and some other models. Since this is a binary task, is there an explanation for that low accuracy?

**Additional Feedback:**

Since NLP and NLU are not just about texts, I would recommend paying attention to tasks and benchmarks on other types of data that fall within the scope.

**Correctness:**

Personally, I don't think there is a need to emphasize "we do not include a separate validation set for any tasks". As long as the dataset is providing a separate public/private test set, validation is a part of the training process and a strict validation split is rarely used in the latest approaches. Cross-validation is a very good substitute, which is decided by the ML engineer.

**Documentation:**

OK

**Ethics:**

OK

**Relation To Prior Work:**

OK

**Summary And Contributions:**

The paper emphasizes existing dataset limitations that undermine claims of achieving broad human-level performance on NLU tasks. The main point is that previous works have been trained on a large corpus of data, which is not similar to the human learning process, thus comparing them would not be valid.

Based on this premise, the paper presents benchmark few-shot datasets for 6 NLU tasks and compares results of existing models on them.

---

> ### Author Response · Authors · 2021-09-28
> **Author Response**
>
> We thank the reviewer for the helpful feedback. Please find our response below.
>
>
> ### 1. Test Sets
>
> We sample the test data (~200 samples for each classification task and for each question type for the MRC/NER tasks) from the original test set available for every task (refer to Section 3.2, Lines 168-191). One of the major objectives of this work is to demonstrate human-machine performance gap for few-shot settings. To this end, we need to compare human-machine performance on the same test split for every task. Since it is very expensive to obtain multiple human annotations (to measure agreement) on thousands of test examples, we perform sub-sampling to create smaller test sets.
>
>
> ### 2. Variance of BERT on CoNLL/WikiANN
>
> As opposed to instance-level classification tasks, NER tasks (e.g., CoNLL and WikiANN) require token-level annotations and correspondingly are much more challenging in the few-shot setting. The token-level annotations (e.g., B, I, O) indicate whether a token is part of the answer or not. In this particular case, Bert-Base marked all the tokens as ‘O’ indicating out-of-span, thereby, generating empty lists as answers. The default accuracy indicates the proportion of empty lists (questions that do not have answers) in our test sets and hence the 0 variance.
>
>
> ### 3. Low Accuracy of Some Models
>
> For classic finetuning, an additional task-specific layer with thousands of randomly initialized parameters is required. This turned out to be quite hard to learn with few samples for some models showing very low accuracy and often very high variance (e.g., 21.7 for RoBERTa-Large and 11.9 for DeBERTa on SST-2 when K=10).
>
>
> ### 4. Beyond Text
>
> While we agree that NLP can be about more than text, we opted for focusing on text for this work. We hope to expand to other dimensions in future work.

---

### Author Response · Authors · 2021-09-28
**Summary of Key Responses to All Reviewers**

We thank all the reviewers for their helpful feedback. We will include all the suggestions and our responses in our revision. We would like to highlight some important distinctions and maintenance plan for our benchmark as follows:


### 1. Related Work on Few-shot Learning and Existing Benchmarks in NLU

CROSSFIT focuses on multi-task and meta-learning settings where the models have access to data from many training tasks to learn from, in order to evaluate the few-shot learning ability on new unseen test task. This is different than CLUES which does not address the multi-task setting. Rather, CLUES consists of a carefully chosen set of fixed tasks, each with its own (small) training set and test set.

Few-GLUE and Few-CLUE were both created by sampling 32 instances from each task in GLUE and CLUE respectively. We summarize the difference between CLUES and the other two datasets below:

|                                        | CLUES                                     | Few-Glue        | Few-Clue             |
|----------------------------------------|-------------------------------------------|-----------------|----------------------|
| Language                               | English                                   | English         | Chinese              |
| Tasks                                  | Classification, MRC and Sequence tagging  | Classification  | Classification, MRC  |
| Few-shot Splits                        | Multiple                                  | Single          | Multiple             |
| # few shots                            | 10, 20, 30                                | 32              | 32                   |
| Unified Format                         | Yes                                       | No              | No                   |
| Unified Metric                         | Yes                                       | No              | No                   |
| Few-shot Human Performance Evaluation  | Yes                                       | No              | No                 |

In summary, CLUES: (1) expands task coverage to include MRC and sequence tagging in addition to classification, (2) provides multiple few-shot splits and sizes, (3) provides a unified format for all tasks to eliminate task-specific model customization, (3) provides a single unified metric across all tasks and (4) provides results for both human and machine performance on all few-shot tasks

### 2. Multimodality

While we agree that multimodal understanding is an interesting direction, our focus in this work was limited to natural language understanding rather than multimodal understanding. As such, in our human evaluation, we consider a closed-world assumption and provide the individuals with only the textual information for every task.

### 3. Code Release

The implementation of all baselines, evaluation scripts, sampling and data processing scripts etc. will be made publicly available on Github. The code and data are available for review in the following link: https://aka.ms/CLUES-Code

The data has also been made available during the review process in the submitted URL.


### 4. Maintenance Plan

We will maintain a leaderboard in our Github page, allowing researchers to submit their results as entries.

#### Submission Requirements:

- Each submission must be submitted as a pull request modifying the markdown file underlying the leaderboard.

- The submission must attach an accompanying public paper and public source code for reproducing their results on our dataset.

- A submission can be towards any subset of tasks in our benchmark, or towards the aggregate leaderboard.

- For any task targeted by the submission, we require evaluation on all 5 splits of the corresponding dataset and a report of their mean and standard deviation.

#### Maintaining and Evaluating Submissions: We will check each submission, to the best of our ability, that:

- The model did not pre-train directly on our dataset.

- The submission does not use external data or data from other splits during few-shot fine-tuning, either as extra training set or as validation set for hyperparameter tuning.

- There are no other obvious errors. However, we will not evaluate the merit of the submission’s technical approach.

- We will attempt to reproduce a submission’s results only if we believe there is an error or the submission is made in bad faith.

---

### Decision · Program_Chairs · 2021-10-09

**Decision:**

Accept

**Comment:**

This paper presents a benchmark dataset for few-shot learning based on the GLUE benchmark for natural langauge understanding. While there exist previous benchmark datasets, this work presents a new benchmark with significant novel contributions over the previous datasets. This would be a useful resource for NLP researchers.